# Do We Need Public Green Spaces Accessibility Standards for the Sustainable Development of Urban Settlements? The Evidence from Wrocław, Poland

**DOI:** 10.3390/ijerph20043067

**Published:** 2023-02-09

**Authors:** Justyna Rubaszek, Janusz Gubański, Anna Podolska

**Affiliations:** Department of Landscape Architecture, Wrocław University of Environmental and Life Sciences, 55 Grunwaldzka Street, 50-357 Wrocław, Poland

**Keywords:** green spaces provision, standard approach, proximity of public green spaces, planning of residential areas, housing estates, neighbourhood, densification, urbanisation

## Abstract

Public green spaces (PGSs) have a positive impact on the health and wellbeing of urban residents. However, their accessibility can be limited due to high urbanisation pressures and inadequate or insufficient regulatory provisions. This problem has been recognised for Central European cities, of which Wrocław is an example, where the provision of PGS accessibility has not received major attention in recent decades, and the planning system has been in constant transformation since the transition from a centrally planned to a free-market economy. This study therefore aimed to examine the distribution and accessibility of PGSs in the expanding area of Wrocław today and upon implementation of the plans under the proposed standards. These analyses were performed using the QGIS application, network analysis, and the ISO-Area as polygon algorithm. The findings revealed a conspicuous lack of available PGSs, which includes areas above 2 ha, such as district and neighbourhood parks. New PGSs are now being planned, but even so, part of the residential areas will remain outside their catchment zone. The results obtained provide strong evidence that it is essential that standards should be a tool implemented in urban planning, and that the adopted procedure can be transferred to other cities.

## 1. Introduction

In view of the scenario forecast that by 2050 sixty-eight percent of the world population will live in cities [1], planning of liveable and healthy urban settlements whose inhabitants have guaranteed access to public green spaces is one of the vital roles to be played by planners and public administrations. Even more so as the need to provide accessible, safe, inclusive green spaces is explicitly voiced in the 2030 Agenda: Transforming our world: the 2030 Agenda for Sustainable Development [2].

Indeed, green spaces, including public green spaces (PGSs) such as squares, parks, and other green areas open to everyone, enabling recreation and leisure for urban residents are extremely important for people’s mental and physical health and wellbeing [3,4,5,6,7,8,9,10]. Exposure to greenery reduces stress and supports psychological restoration [11], encourages physical activity [12,13], enables social interaction and it can improve social cohesion [14,15,16,17,18,19]. The frequency of using PGS is greatly influenced by their close proximity [20,21,22,23]. Proximity is also one of the most important factors, next to, e.g., the attractiveness of the area, that determine the improvement of the wellbeing among its users [19,21,22]. Sturm and Cohe [21] in their study found that the highest level of mental wellbeing was achieved among people living within a short walking distance from the park (400 m) and decreased significantly over the next distances. Users who live near parks also report better physical health and life satisfaction, tend to develop greater social interaction and perceive it as a more secure location [22]. Proximity to PGS is therefore considered to be a key indicator of their accessibility [24]. By the same token, the PGS are ‘accessible’ when they exist within a suitable distance from the place of living and when we feel that we are welcome there, and we can freely reach them and safely use them [25]. 

One tool to support planning for the accessibility of green spaces, in particular PGS, is the existing standards. The need for their use was noted by many authors analysing the accessibility of PGSs in various spatial and social contexts [26,27,28,29,30,31,32]. Standards define maximum distances to particular types of green spaces, which differ in size and therefore in their capacities for providing appropriate facilities and amenities [26,27,28]. Larger green areas allow for more activity, including walking for many users [33,34]. Smaller green sites located close to the home, on the other hand, are important for the attractiveness of the neighbourhood, are an incentive for many people to go out of the house, have a positive effect on ‘socialising’ and ‘rest and restitution’ [35,36]. There is no single predetermined distance to particular types of PGS and, there is no fixed site size and associated distance [24]. However, the distances defined in the pedestrian accessibility standards for PGS are similar, as they derive from the real possibility to cover a certain distance on foot. The closest distance, a distance of 5 min walking, is adopted as 300–400 m [24,26,28].

Since the beginning of the 20th century, there has been an increase in research on the accessibility of green spaces and PGS in Western European countries [26,29,37,38,39,40]. In recent years, this issue has also become the subject of research in Asian countries [28,32,41,42,43] and in selected cities in South America [44] and the Middle East [45]. The accessibility of green spaces has also become a subject of research in selected cities in South America [44] or the Middle East [45]. The hitherto studies have examined accessibility for the general population as well as for groups with different social and economic status, e.g., ethnic and religious minorities [37,46], and other groups for whom accessibility of green spaces is limited due to age, mobility, gender constraints, e.g., the elderly and women [41,44,47,48]. Relatively few studies have examined pedestrian accessibility at the neighbourhood scale from the perspective of sustainable development of residential areas. Moreover, when pedestrian accessibility was studied [28,49,50,51], the assessments were concerned with the current situation and did not take into account changes resulting from the planned development.

As far as the situation in Central Europe including Poland is concerned, the issue of accessibility of green spaces has been mostly disregarded in spatial planning and far from being adopted as a guiding principle for urban greening [52]. In Poland, where the change from a centrally controlled economy to a free-market economy took place in the early 1990s, and the entire planning system was successively transformed, no sufficiently effective tools have yet been developed to ensure sustainable urban settlements that cater for public health and environmental justice in access to PGS. Meanwhile, the apparent development pressure looms large as one of the main threats to the current but also future accessibility of green spaces [53,54]. The more so as residential development is primarily based on private, developer-driven investments, oriented towards profit [55], and this results in, among others, insufficient provision of greenery [56]. 

In the development of Polish cities, attempts are being made to follow the compact city model [57]. However, this model, despite its many positive attributes, also carries risks with regard to access to green spaces [58,59,60]. As Grêt-Regamey et al. noted, “while compact cities aim to reduce land consumption, densification puts pressure on the remaining green areas, influencing ecosystem services provision and ultimately the life quality of the growing urban population” [61]. 

Spatial development in Poland is implemented on the basis of land use plans. Land use plans are established on the Study of the Conditions and Directions of Spatial Development, a form of strategic document prepared for the entire municipality. The only regulations regarding greenery and green spaces are those concerning the biologically active areas within the boundaries of the plot concerned [62]. With investment pressure, there is a danger that developing areas of Polish cities will be deficient in greenery, especially PGS, the planning of which is not supported by either regulations or standards.

With these concerns in mind, the aim of our study is to analyse the pedestrian accessibility of PGS in the developing parts of Wrocław, a medium–large city located in southwestern Poland. Wrocław is an interesting case study because it represents cities form Central Europe that have been developing dynamically since the political and economic transformation of the 1990s, but in which ensuring the accessibility of PGS was not a key issue in greenery planning. In Wrocław itself, there are no regulations supporting the accessibility of PGS, which, given the high urbanisation pressure, poses a threat to the quality of life in the city, and thus the health and wellbeing of residents. With this in mind, we proposed PGS pedestrian accessibility standards, which are particularly needed in the context of sustainable development of new housing estates. Based on the standards, we established the current accessibility of PGS and whether, and how, this will change in the future along with the progressive development of residential areas and implementation of land use plans. 

Embarking upon this exploratory study, we assume a hypothesis that the absence of the application of standards in local planning limits the pedestrian accessibility of PGS in developing residential areas of the city and that PGS are not created parallel to the built-up sites and infrastructure, and if they are, they are created secondarily, at a later stage.

## 2. Materials and Methods

### 2.1. Study Area

The study was carried out in Wrocław, a city located in southwestern Poland (51° N, 17° E), in Europe. Wrocław is the fourth largest city in the country, it covers an area of 29,281.65 ha (according to data from the Local Data Bank, as of 12 April 2021) and has 642,700 inhabitants [63]. Approximately 31.40% of the city area comprises built-up land, 28.90% agricultural land (cultivated and bare fields), 36.6% urban green and open spaces, and 3.10% water [64]. Wrocław is a dynamically developing city in economic terms, which manifests itself, among other things, in the development of housing; between 2017 and 2020 housing growth was 9.8%, the highest among Polish cities. Currently, the city is ranked 4th in Poland in terms of housing stock with 351,000 dwellings [65]. These trends are also recognised on a European scale; according to the Deloitte report [66] in 2020, Poland had the highest number of flats completed in Europe per 1000 citizens.

Wrocław adheres to the concept of a compact city and aims to increase the size of residential areas within its borders and limit urban sprawl. The document indicating the directions of spatial development is Study of Conditions and Directions of Spatial Development (spatial development study), and the basis for the implementation of individual investments are land use plans drawn up for individual fragments of the city. At the end of 2021, 492 local plans were in force in Wrocław, they covered an area of 17,798.29 ha, representing 60.8% of the city’s space [65]. Since 1991, Wrocław has been administratively divided into urban settlements, which are subsidiary units of the municipality. Individual settlements have their own budget and representatives in the city council. The current boundaries of the districts date from 2016 when they were last revised. 

In 2019, the city launched the ‘complete neighbourhoods’ programme; in the city’s vision it is the housing estates that are to become sustainable healthy neighbourhoods. The programme involves working with residents and diagnosing their needs on several key issues, one of which is: “movement, leisure, health” provided by “neighbourhood greenery, parks, squares, open recreational areas, small-scale walking and cycling routes” [67]. So far, four of Wrocław’s housing estates have been included in the programme. Among them is the Jagodno estate, one of the three areas included in the study. The outcome of the public consultation reveals, unfortunately, a poor provision of accessible parks and a lack of space to spend time together outdoors [67,68]. The other areas included in the study are the neighbouring districts of Ołtaszyn and Wojszyce. For the purposes of the study we labelled them (I) Ołtaszyn, (II) Wojszyce, and (III) Jagodno (Figure 1). 

The estates form a developing urban unit located in the southern part of the city, bordering on suburban zones. It covers an area of 1006.54 hectares, which accounts for 3.44% of Wrocław’s total area. The area of the contemporary districts was incorporated into the city limits in 1950, when the territory of Wrocław was enlarged for the last time. Wojszyce and Ołtaszyn were historical villages that had been established as early as in the Middle Ages, while Jagodno was a colony of Brochów, a suburban settlement of Wrocław, and an independent town in the years 1939–1950 [69]. In the 1970s and 1980s Wojszyce and Ołtaszyn, in the vicinity of the historical village buildings, complexes of detached, semi-detached, and terraced houses began to emerge. The dynamic development of Jagodno began at the turn of the 20th and 21st centuries. A brief overview of the neighbourhoods and the area they occupy is presented in Table 1.

There are many open spaces around the built-up areas; these are mainly agricultural land, including arable fields: 275.79 ha; grasslands: 229.91 ha; and permanent crop land: 126.1 ha, including allotment gardens: 101.58 ha; orchards: 23.52 ha; and ornamental plant nurseries and plantations: 1 ha. Only 19.2 ha, is occupied by wooded areas [70].

### 2.2. Socio-Cultural Characteristic of Residents form the Study Area

The residents of the study area range from the elderly to families with children. Seniors live mainly in single-family buildings from the 1970s and 1980s, and people of working age live in the newly erected multi-family and single-family buildings. They are mainly people of Polish nationality, and of medium and high social status. Residents are actively involved in matters relating to their surroundings; for example, as part of the Wrocław Civic Budget, they submit proposals for various investments, including those related to the creation or renewal of green spaces. The inhabitants’ needs with regard to spending time in the outdoor space were partly recognized thanks to the “complete estates” programme (see Section 2.1). As part of this initiative, the residents of Jagodno noted the scarcity of parks in their neighbourhood where they would like to be able to relax and also spend time actively, as well as have the opportunity to meet and better integrate with their neighbours [67,68]. To date, no in-depth research has been performed on the culture of using outdoor spaces, including PGS in the study area as well as in the city.

### 2.3. Data

We used the following publicly available data to conduct the study:-Publicly accessible database of topographic objects BDOT10k [70];-Publicly accessible orthophotomap 2021 made on the basis of digital images acquired in the 2nd–3rd quarter of 2021, with field resolution of 25 cm/pixel [71];-Network and street axes from OpenStreetMap^®^ licensed under the Open Data Commons Open Database License (ODbL) by the OpenStreetMap Foundation [72];-Municipal data on urban greenery retrieved from Wrocław Planning Office, city unit responsible for planning and spatial development of the city;-Study of the Conditions and Directions of Spatial Development of Wrocław from 2018 [73];-Land use plans prepared for Wrocław [74].

### 2.4. Methods

The flowchart of this study is as follows (Figure 2).

We divided the research into two main stages: the first one was to analyse the pedestrian accessibility of PGS now; the second one to analyse them after the implementation of the land use plans of the studied housing estates. 

In the first stage of the research we identified PGS. To perform this, we used the data on urban green spaces provided by the Wrocław Planning Office and data contained in the Study of the Conditions and Directions of Spatial Development 2018 [73]. As PGS, we considered, in line with other researchers [9,28,33,36,48,49], those areas that can be used by residents without restriction and have been deliberately planned as places for recreation and leisure or have been secondarily adapted to perform these functions. This group of areas includes squares, playgrounds with greenery, pocket parks, parks and urban forests. We did not include areas such as allotment gardens as PGS, due to the fact that they cannot be used by all residents, despite the fact that under Polish law, according to its formal definition, they are classified as PGS [75]. We also did not investigate the current accessibility of open spaces as these areas which, although an important part of the city’s green infrastructure and provide many cultural ecosystem services [76], have not been officially designated for recreational and leisure functions. Instead, we were concerned with whether and how the designation of open spaces change on the basis of spatial development plans, whether open spaces will be preserved or transformed, and if so, whether some of them will be allocated to PGS. This was the focus of Stage II of the research.

We classified the PGS into: PGS I, with an area of <2 ha; PGS II, 2–5 ha; and PGS III, with an area of >5 ha and by applying the standards we proposed for Wrocław we determined the maximum walking distance to each PGS category: 400 m, 800 m, and 1200 m (Table 2).

The 400 m (1/4 mile) baseline distance adopted in the standards is found in many other standard-based PGS accessibility studies (cf. [26,27,28]). The distance a person is prepared to cover on foot, in about 5 min, is 400 m [24]. The 400 m range has determined the location of basic services, public transport, and green spaces since C. Perry’s neighbourhood unit concept was conceived [77]. Today, the 400 and 800 m distances are used in sustainable neighbourhood and city planning (cf. [78,79]). In the 400 m range, the smallest green spaces are planned to be located with a basic programme most often corresponding to the needs of children and the elderly. Within 800 m, many standards, including those in the 1970s and 1980s promoted in Poland [80] stipulate the location of PGS with an area of at least 2 ha. This is the minimum surface that is considered to allow residents to carry out various activities [26,37]. At further distances, increasingly larger parks should be planned to serve several neighbourhoods, a district, and then the whole city. In planning the hierarchical structure of the PGS, it is assumed that the smaller areas satisfy the basic, i.e., daily needs for leisure and activity in the green space. The larger areas, to which people are willing to travel longer distances using different means of transport to reach, are frequented at weekends or more sporadically. The pedestrian accessibility of PGS is particularly important for shaping healthy and liveable neighbourhoods, where walking is seen as a viable and desirable means of transportation [78].

During Stage I of the study, we also identified built-up areas and, of these, we specified residential areas, dividing them into those with single- and multi-family housing; as it is with reference to these that we analysed the accessibility of the PGS in the next steps. 

We made the identification of the residential areas based on publicly available data: the 2021 orthophotomap [71] and the BDOT10k database, from which we obtained the registered boundaries of the plots [70]. We used QGIS for data analysis and graphical presentation.

In the next step, we investigated the accessibility of the PGS with the application of network analysis and the algorithm Iso-Area as Polygon in QGIS in the QNEAT3 plug-in, the QGIS Network Analysis Toolbox. Using the centroid tool, we determined the central points of PGS I, from these points we made network analyses. For PGS II and III, we performed a network analysis not from the central points, but from the entrances, due to the larger size of these sites. We took as entrances the points of intersection of the site boundaries with the axes of the streets and the axes of the footpaths. We did not include highways or motorways because, as a rule, pedestrian traffic is not allowed within them, and they also did not exist in the study area. The network of streets and footpaths for the network analysis was obtained from Open Street Maps (OSMs) [72]. After performing a network analysis for the individual distances, we applied the algorithm Iso-Area as Polygon in QGIS, which allowed us to determine the services areas of PGS. Within the service areas there are lands with various functions, among them we distinguished sites of residential development, dividing them into single-family and multi-family housing areas (Figure 3).

While studying the accessibility of PGS, we assumed that larger sites can take over the function of smaller ones, so for the distance of 400 m we performed accessibility analyses for three categories of sites, i.e., PGS I, II and III, and for a distance of 800 m we considered two, PGS II and III, i.e., sites of 2–5 ha and >5 ha. When examining the 1200 m access range, we already included only sites of >5 ha, i.e., PGS III. This has not been adopted so far in other standards-based accessibility studies.

In phase II of the study, we investigated whether and how the accessibility of the PGS would change if the land use plans were implemented. On the basis of the land use plans, we specified the future distribution and area of the PGS and residential areas, and then repeated all the steps of analysis, in the same manner as in Stage I.

As a result of our analyses, we identified areas of residential development located within and outside of the individual PGS ranges. Ultimately, we aggregated the accessibility ranges and designated areas of residential development located outside of all PGS services zones. Residents of the areas so designated do not have access to any of the existing or planned PGS. 

## 3. Results

The Stage I analyses showed that there are nine PGS I (pocket parks, green squares with playgrounds), three PGS II (neighbourhood parks), and five PGS III (supra-neighbourhood parks) within the study area of 1006.54 ha, and within the relevant buffer zones at distances of 400 m, 800 m, and 1200 m from the border of the study area (Figure 4). 

All PGS I are located within the boundaries of housing estates; only one PGS II is located within the boundaries of housing estates, the others in the buffer zone of 800 m; and all PGS III are located outside their boundaries, in the buffer zone of 1200 m. Representative PGS types are shown in Table 3.

As for the sites with residential development, they cover 33.72% of the study area (339.44 ha), of which 76.20% (258.6 ha) is single-family housing and 23.80% (80.84 ha) is multi-family housing. PGS I cover a total of 3.99 ha, which represents only 0.1% of the current residential areas. PGS I are located in the estates: Ołtaszyn (1) and Wojszyce (2). Two of the nine PGS I are former village squares, the other four were created together with single-family housing already in the 1980s, and three were newly established. As for the total area of PGS II, it amounts to 8.33 ha, which accounts for 2% of the residential areas. One of the PGS IIs is a remnant of the historical greenery complex, the others were created in the second decade of the 21st century. PGS III covers a total of 71.22 ha, which represents 17.82% of the current residential areas of the housing estates. PGS III are located in a buffer of 1200 m; four of them are historical green spaces (parks, gardens, and former cemeteries converted into parks), the fifth is a modern establishment, located among multi-family large-panel housing from the 1980s (Appendix A: Appendix A). The distribution of the PGS in relation to the residential sites with single and multi-family housing is shown in the Figure 5.

Based on a network analysis and the Iso-Area as Polygon algorithm in QGIS, we examined how the accessibility of PGS is currently developing. There are 28.9% (98 ha) of sites with single-family housing and 3.73% (12.68 ha) of sites with multi-family housing: a total of 32.65% (110.68 ha) of residential land is within 400 m walking distance of all three PGS types. Outside this range, 67.40% of the sites (228.76 ha) are located: 47.13% (160 ha) with are single-family housing and 20.27% (68.16 ha) are multi-family housing. 

Within the walking distance of 800 m to PGS II and III, there is 11.2% (37.96 ha) of land with single-family housing and 0.8% (2.71 ha) with multi-family housing, and outside the reach of 65% (220.04 ha) and 23% (78.13 ha) respectively, for a total of 88% (298.77 ha) of land outside PGS II. Within the walking distance of 1200 m to PGS III, 11.45% (38.83 ha) of residential land is located, of which 10.54% (35.80 ha) is single-family housing and 0.9% (3.03 ha) is multi-family housing. Outside the 1200 m range are 65.5% (222.20 ha) of single-family development sites and 23.06% (77.81 ha) of multi-family development sites, for a total of 88.57% (300.58 ha) of residential sites are areas with a PGS III accessibility deficit. The summary results of the accessibility analysis are presented in Table 4 and in the Figure 6 and Figure 7.

The aggregation of the services areas revealed that currently 54.3% (184.32 ha) of the residential development sites are located outside the ranges of pedestrian access specified in the standards, of which 35.3% (119.69 ha) are single-family housing and 19% (64.63 ha) are multi-family housing development (Figure 8).

The results of the second stage of the study, demonstrate that the land use plans provide for an increase in built-up land, which leads to the spatial amalgamation of estates and their resulting occupation of the current open spaces. In the place of the current open spaces, apart from the housing development, PGS is also planned (Figure 9).

According to the land use plans, the area of residential development will increase by 63.60% and will amount to 555.53 ha, of which single-family development will occupy an area of 415.65 ha, an increase of 60.70%, and multi-family development will amount to 139.88 ha (an increase of 73%). The differences in the area of residential land between the current and planned state are shown in the chart (Figure 10). Single-family development will continue to be the predominant residential development type accounting for 74.8% of all planned residential sites.

The surface area of PGS according to land use plans will also increase and will total 174.33 ha (an increase of 108.7% (90.79 ha)). The size of the land allocated to PGS I will increase by 197% (7.86 ha); however, it will be quite small at 11.85 ha: the size of the land designated for PGS II will increase by 24.73% (2.06 ha) to a total of 10.39 ha, the size of PGS III by 113.55% (80.87 ha) to a total of 152.09 ha (Figure 11).

Regarding PGS I, fourteen new sites are planned to be created, two of them in the buffer of up to 400 m from the western boundary of the study area. The plans do not provide for the creation of any new PGS II sites, but for the enlargement of one of the existing sites. They do, however, provide for the creation of two new PGS III sites with a total area of 80.87 ha, one of which will be located immediately outside the western boundary of the study area and the other between the neighbourhoods (2) Wojszyce and (3) Jagodno. The first one will be transformed into a park, the second into an urban forest with paths and rest sites. The current as well as the planned distribution of PGS are illustrated in Figure 12.

Implementation of the land use plans aims to improve the accessibility of the PGS at each of the ranges adopted in the standards, yet there will be part of the residential sites outside of the PGS service areas. There will be 47.94% (266.31 ha) of residential sites within the walking distance of 400 m to PGS I, II, and III, and 52.06% (289.22 ha) outside of it. Within the 800 m walking distance to PGS II and III there will be 253.86 ha or 45.7% of residential sites and 301.67 ha or 54.3% outside of it. A total of 403.86 ha (72.7%) of residential land will be located within the 1200 m range and 151.67 ha (27.3%) outside of it. The results of the analysis of PGS accessibility within each range are summarised in Table 5, and shown on the Figure 13 and Figure 14.

Upon aggregation of the service areas, 16.61% (92.29 ha) of residential development will be found outside the established ranges of pedestrian accessibility (Figure 15).

This accessibility deficit will affect both residents in multi-family housing located in an area of 30.30 ha (21.66% of all multi-family housing and 5.45% of all residential areas) and residents in single-family housing in an area of 61.99 ha (14.91% of all single-family housing and 11.16% of all residential areas). Despite the fact that this accessibility will be better than at present, some of the residents will be deprived of access to PGS within the established ranges (Figure 16).

## 4. Discussion

### 4.1. Discussion of the Research Procedure and Results

Like many other authors analysing the pedestrian accessibility of PGS [28,29,37,42,81,82,83,84,85,86,87], we applied GIS application. Based on network analyses and the Iso-Area as Polygon algorithm, we determined PGS service areas. PGS service areas calculated thus are more accurate than in the other popular method for studying the accessibility of green spaces, which is the buffer zone method, based on linear distance to generalized serviceable areas around facilities [28,29,81]. The network analysis made it possible to determine the road network with the length corresponding to the assumed pedestrian access distances, the use of Iso-Area as Polygon algorithm to determine the area of the impact of the PGS. The combination of the two analyses provided the opportunity to verify whether the area determined by the algorithm really overlaps with the road network.

In our research, as in the many other studies of accessibility of PGS [32,37,42,81,82,83,84], we used municipal data provided by the planning office. The results obtained on their basis are credible for municipal administration units and thus can be more willingly used when making decisions on the development of urban greenery. PGS identification can also be carried out on the basis of generally available data: orthophotomaps, land use, and land cover data from Urban Atlas [28,29,86] or on the basis of other data, for example the database of topographic objects (BDOT10k) developed for Poland [88]. However, PGS identification based on orthophotomaps and/or land use and land cover is time-consuming, and can be difficult without additional information on PGS, especially when the research involves the entire city. 

Our analysis of the accessibility of PGS involved taking into account the surface area of the study sites and their hierarchy, which means adopting the assumption that as the surface area of PGS increases, so does their range of influence, capacity, and the possibility of implementing a larger, more extensive programme. An approach that takes into account the hierarchy of green spaces appears in standard-based studies of accessibility analyses [26,27,28,32,37,84]. However, given that PGS should be planned with a hierarchical structure in mind, we adopted a new assumption that larger sites, located closer to the maximum range set by the standards, take over the function of smaller sites. This approach is more flexible than the ones used so far and, and we believe it facilitates PGS planning in relation to local conditions. With the assumption that larger PGSs acquire the functions of smaller ones, their development should be concerned with the distribution of facilities and amenities intended for children, young people, or the elderly in such a way that they are located at a convenient distance, not exceeding 400 and 800 m, from residential areas.

Also important in our study was the inclusion of PGS located outside the boundaries of the study area, within the standardised distance buffers. However, the buffers did not include peri-urban areas, which as in other studies (cf. [37]), was justified by the use of urban datasets, but also by the need to translate the results into the actions of the city administration and urban planners. Not including PGS located outside the boundaries of the study area may distort the results to a certain extent, we can imagine a situation where a rural park located close to the urban boundaries provides a place for recreation for city residents. 

Due to the different data used for the study and some differences in methodology, e.g., considering only areas larger than 2 ha [37,87], comparing our results with other findings is somewhat difficult, although there are some common reports to be noted.

There is a noticeable paucity of green spaces located within a 5 and 10 min walking distance (300–400 m and 800 m) and an improvement in accessibility with increasing distance [32,37,86,87]. This coincides with our results from the Stage II of the study, which also show low accessibility of sites of <two hectares and two–five hectares and better accessibility of larger sites. This means that the compact development layout proposed in the plans, without green areas woven into it, does not represent a good model for sustainable neighbourhoods. Insufficient PGS I with an area of up to 2 ha and PGS II with an area of 2–5 ha included in the development plans is bound to limit residents’ recreation in green spaces within a 5–10 min walk. 

In turn, the best current accessibility of small squares is due to their even distribution among existing older housing developments. These squares were planned as an integral part of the development complexes of the 1980s and 1990s, i.e., before the economic marketization and at the beginning of this process, when the developers’ market was only just forming and the demand for land was incomparably lower than today. 

In each of the two situations that we examined (now and after the implementation of the plans), there is a clear shortage of PGS of 2–5 hectares. This result is in line with the conclusions of the study by Wysmułek et al. [87], who, analysing the availability of PGS in large Polish cities, including Wrocław, showed the greatest shortage of areas with a minimum of 2 ha located within a 5-min walking distance. Our research also confirmed the reports of Quatrini et al. [86] that: “the new is not always better than the old”. Their research showed that the percentage of accessible green spaces is much lower in newer settlements of Rome (Italy), especially those located in the more peripheral administrative units of the city. Regrettably, the law in force in Poland since the 1990s, i.e., since the marketization of the economy, does not enforce the obligation to provide new residential areas with PGS (cf. [79,88]). It only defines the mandatory percentage of biologically active area within the boundaries of the construction plot [62], i.e., in the case of residential areas, on the plot with residential buildings, not outside of it. The surveys we performed highlight this state of affairs thoroughly: there is only one PGS larger than two hectares within the boundaries of the housing estates, the remaining areas of such size are located outside the boundaries of housing estates, in the older part of the city, and only two of them are contemporary neighbourhood parks established after 2000. Meanwhile, it is the sites larger than two hectares only that allow for a more extensive programme and can provide adequate physical activity support for adults [88,89].

### 4.2. Limitations and Prospect for Further Research

The areas designated in the local plans to remain as PGS, for which accessibility analyses were carried out, are those with different ownership structures. Many of them currently do not belong to the city, and in order for them to be officially made available to residents, the issues of sale/purchase and/or lease would have to be settled. This means that the official opening of these areas to residents and their future development may be a long time away. Analysing the issue of land ownership would allow for the presentation of further conclusions regarding the possibilities and limitations of the creation of new PGSs.

Further research can be extended to assess the quality of PGS, as the willingness of residents to use green spaces, and the related health benefits that are also affected by many other factors such as landscape features, equipment, amenities, perceived safety [33,34,35,36,90,91,92,93], biodiversity [94], individual preferences, and attitudes [95]. It may be that some areas are currently too crowded and others less frequented due to lack of appropriate development.

Some of the existing open spaces are informal unorganized green spaces such as greenery along the railroad line and agricultural land lying fallow. As previous studies have shown, these areas can be used by residents for recreation and leisure purposes [76,96] compensating for the lack of PGS, for example [97]. The purpose of our further research can be to identify informal green spaces and examine whether they are used as places for recreation and leisure, and if so, what criteria determine this and whether one of them is the lack of current PGS availability.

Future investigations can also be broadened to include demographic analyses made on the basis of census data to determine how many people live in areas outside the scope of each type of PGS. Such expanded surveys can also provide information on age, gender, occupation, economic activity, household tenure and types, deprivation, disability [25,41,44,47,48], and ethnic minorities [37,46]. For studies of pedestrian accessibility in developing urban areas, it seems particularly important to check the accessibility of PGS for social groups for whom too long a distance may limit or completely prevent the use of PGS; these are especially relevant for children and the elderly [24].

Further research can also include a comparison of the availability of green space in other developing neighbourhoods in Wrocław itself, as well as in other Polish or European cities, and neighbourhoods from different historical periods such as the late 19th and early 20th centuries, the 1920s and 1930s, or the 1970s and 1980s, when other urban development paradigms were in place. The conclusions provided on the basis of such analyses would be very helpful for further discussion and the search for a sustainable form of the city and neighbourhood.

The final point to be discussed is the very implementation of accessibility standards in green space planning. Standards have been debated for, among other reasons, difficulties in enforcing them [60,98,99], and the reliance of urban green planning on the principles of landscape ecology and green infrastructure in response to climate change and environmental concerns [100,101,102,103]. Indeed, it is difficult to implement accessibility standards in densely built-up areas where the amount of vacant land is limited. It is then reasonable to look for alternative ways to increase available green space; for example, in the form of green roofs [104,105,106], pocket parks [107], and green streets [108]. However, the application of standards in developing areas where undeveloped land is available, as our research and that of other authors has shown [28,82,84,85,86], can support a city’s green infrastructure planning and contribute to improving the availability of PGS, thus guaranteeing a better, healthier, and more sustainable living environment for urban residents.

## 5. Conclusions

The procedure of the conducted surveys made it possible to examine the accessibility of PGS in developing urban settlements. Studies were carried out for three ranges of pedestrian access: 400 m, 800 m, and 1200 m, at present and after the implementation of the development plans. Using the GIS application, the PGS service areas were defined, and residential development areas were designated within them. Currently, 32.65% of the residential sites are located within the 400 m service zone of PGS, i.e., the closest pedestrian access distance, and 67.35% are located outside this range. A much smaller percentage of residential land is located in service zones PGS II and III, which are designated by walking distances of 800 m and 1200 m, the percentages are 12% and 11.45%, respectively. The reason for this is that there is one PGS II within the study settlements and all PGS III, or supra-neighbourhood parks, are found outside the settlement boundaries. For the most part, they are historic green space establishments accompanying urban developments. As for the development plans, they provide for an increase in PGS III, i.e., sites of five hectares in size, and associated improvements in accessibility within 1200 m; if the plans are implemented, 72.7% of the residential area will lie within the PGS III service zone. However, there are not enough green spaces provided by the plans within the scheduled compact housing investments, which will limit the accessibility of the PGS within 400 m and 800 m (a 5 and 10 min walk) of over half of the planned development sites. 

An important finding of our research is that PGS in developing urban areas are given minor consideration in relation to buildings and infrastructure, and if they appear, it is only secondary to the latter. Within the boundaries of the settlements under study, which have been developing for almost 30 years, not a single new PGS II and PGS III has been created to date, which marks a deficit of green spaces with a surface area ˃2 ha that encourages walking and enables multiple activities for all groups of residents. Our research therefore confirms that the application of PGS accessibility standards can be an effective support for green space planning in the context of addressing the recreation and leisure needs of residents in developing urban settlements, thereby contributing to the emergence of more sustainable, liveable, and healthy neighbourhoods.

## Figures and Tables

**Figure 1 ijerph-20-03067-f001:**
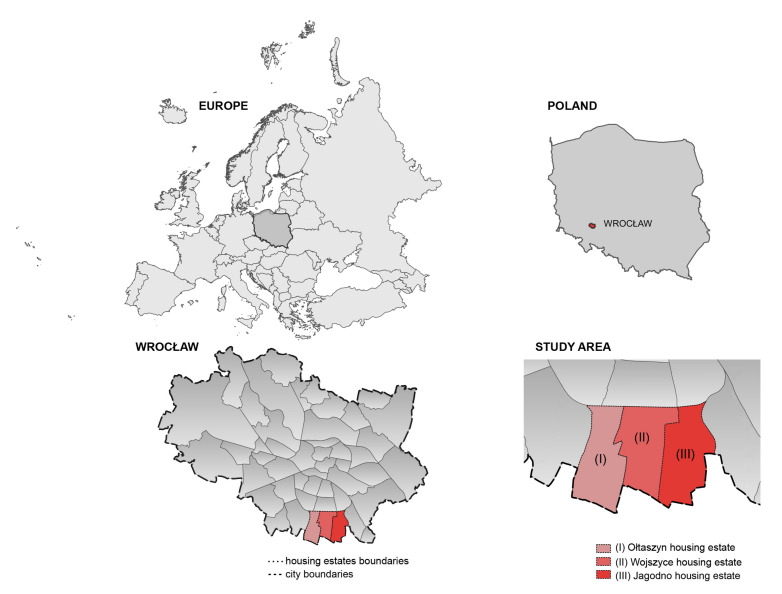
Study area.

**Figure 2 ijerph-20-03067-f002:**
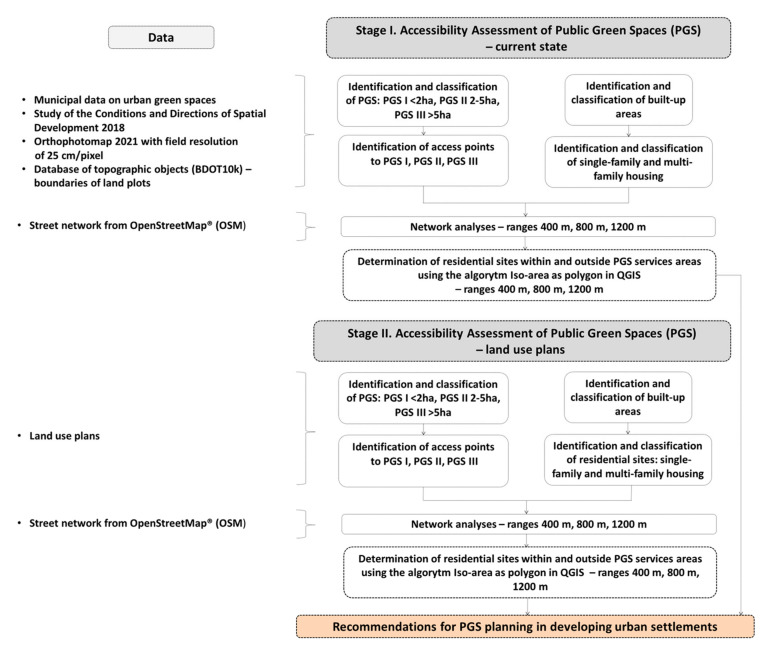
The flowchart of the study.

**Figure 3 ijerph-20-03067-f003:**
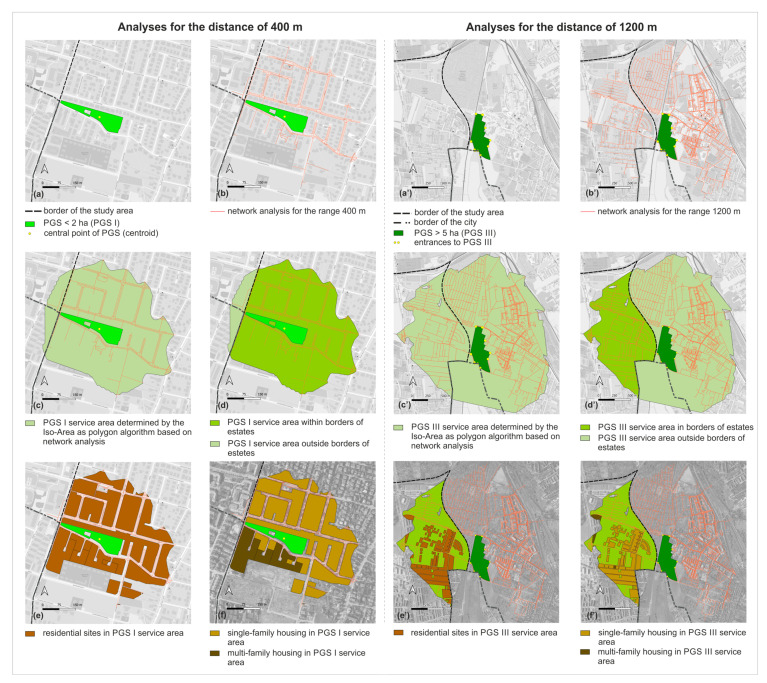
The procedure for determining residential development sites within the service areas of PGS, examples of analyses performed for the walking distance of 400 m and 1200 m. Analyses for the range of 800 m according to the adopted procedure were carried out in the same way as for 1200 m.

**Figure 4 ijerph-20-03067-f004:**
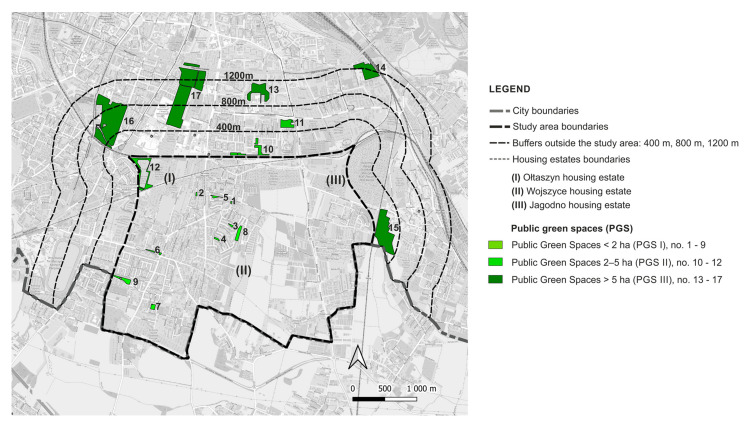
PGS identified in the study area and in buffer zones (400 m, 800 m, 1200 m).

**Figure 5 ijerph-20-03067-f005:**
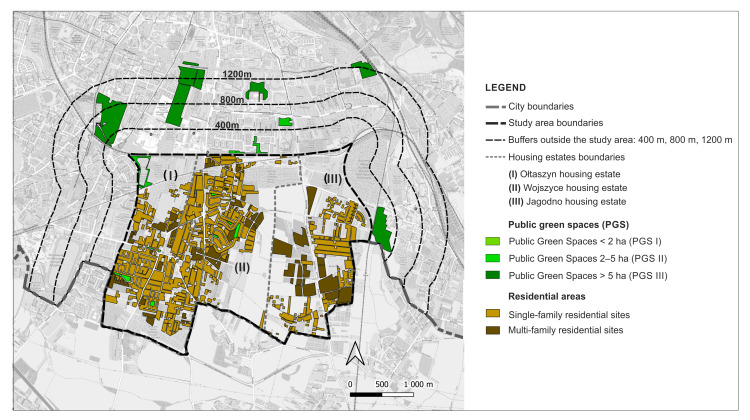
Distribution of PGS and areas of multi-family and single-family housing development: current state.

**Figure 6 ijerph-20-03067-f006:**
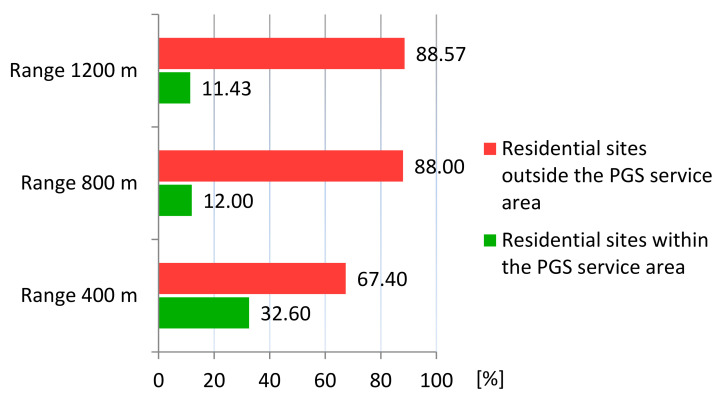
Percentage of residential sites within and outside the service areas of PGS: current state.

**Figure 7 ijerph-20-03067-f007:**
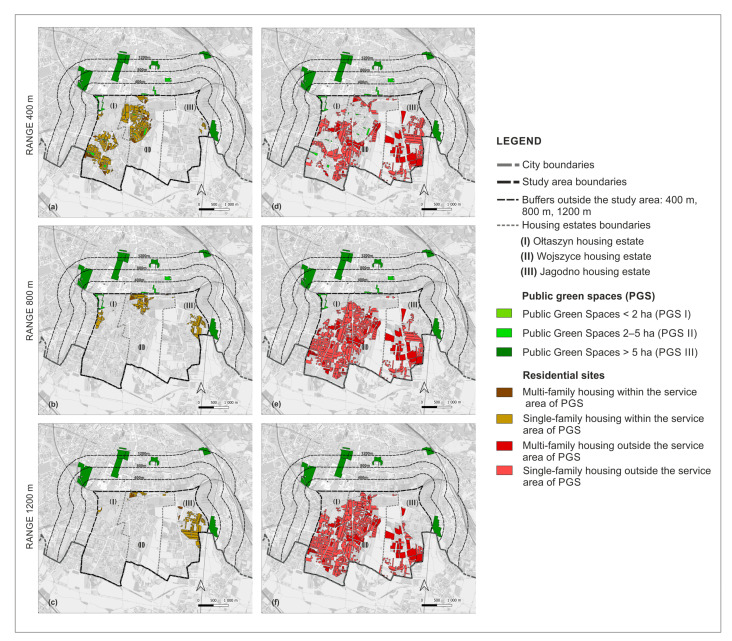
Residential sites within the PGS service areas and outside of them: current state (**a**) Within 400 m; (**b**) within 800 m; (**c**) within 1200 m; (**d**) outside 400 m; (**e**) outside 800 m; (**f**) outside 1200 m.

**Figure 8 ijerph-20-03067-f008:**
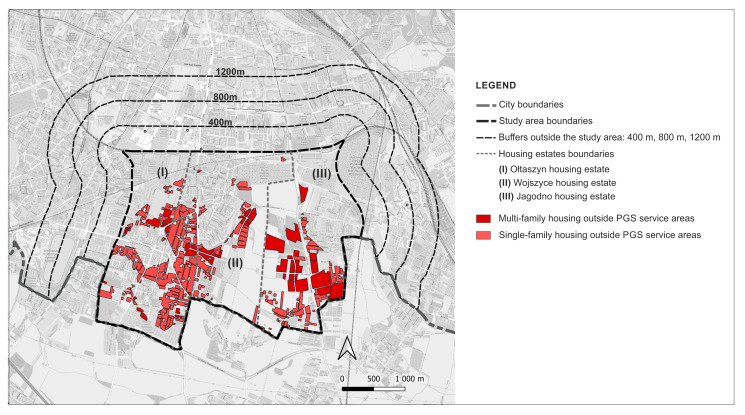
Residential sites not served by any PGS: current state.

**Figure 9 ijerph-20-03067-f009:**
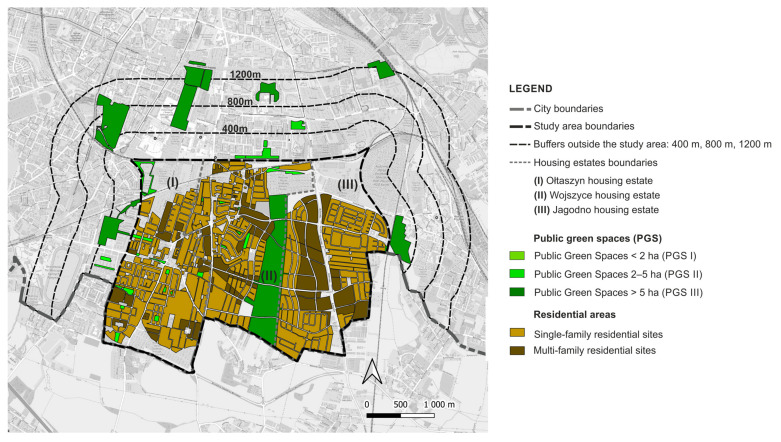
Layout of residential sites and PGS according to land use plans.

**Figure 10 ijerph-20-03067-f010:**
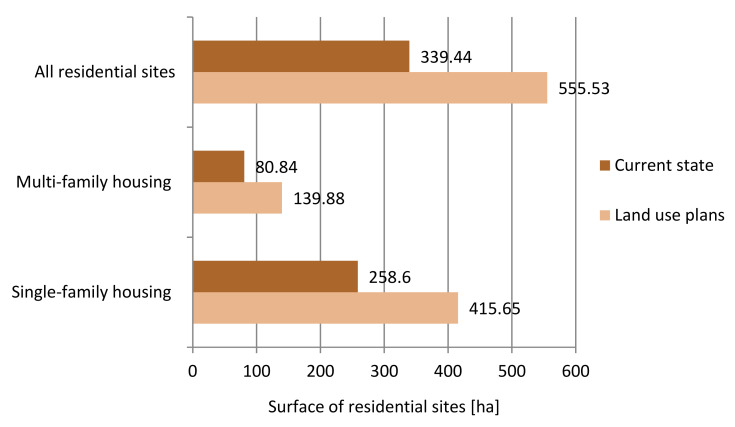
Changes in the area of residential development: current state and land use plans.

**Figure 11 ijerph-20-03067-f011:**
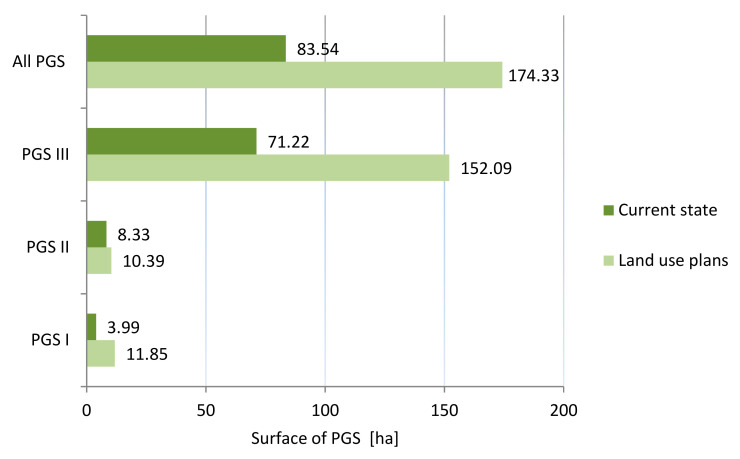
Changes in the surface area of PGS: current state and land use plans.

**Figure 12 ijerph-20-03067-f012:**
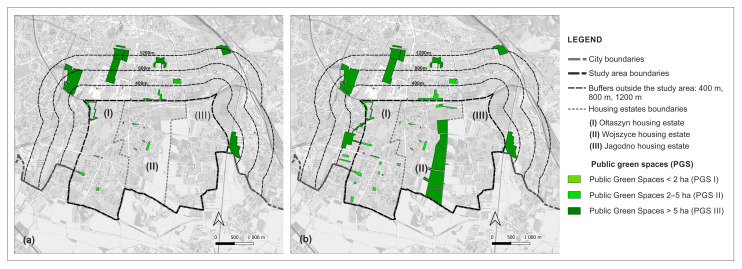
Distribution of PGS in the study area: (**a**) current state; (**b**) land use plans.

**Figure 13 ijerph-20-03067-f013:**
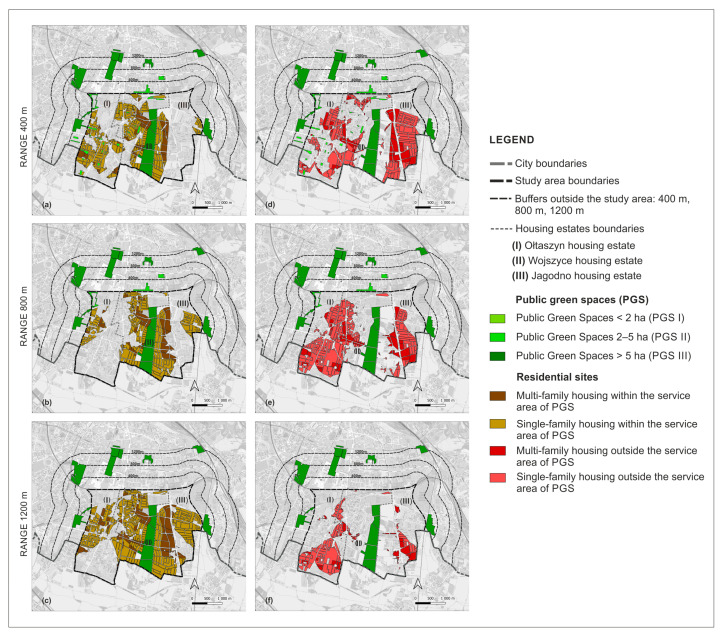
Residential sites located within the service areas of PGS according to land use plans: (**a**) within 400 m service area; (**b**) within 800 m service area; (**c**) within 1200 m service area; (**d**) outside 400 m service area; (**e**) outside 800 m service area; (**f**) outside 1200 m service area.

**Figure 14 ijerph-20-03067-f014:**
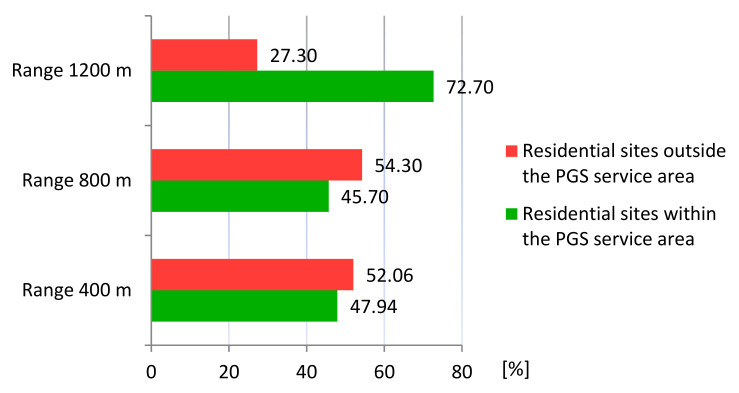
Residential sites in the PGS service areas and outside them according to land use plans.

**Figure 15 ijerph-20-03067-f015:**
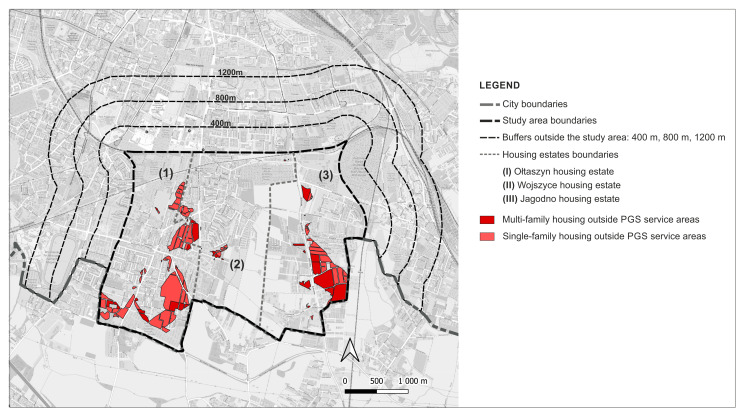
Residential sites not served by any PGS according to land use plans.

**Figure 16 ijerph-20-03067-f016:**
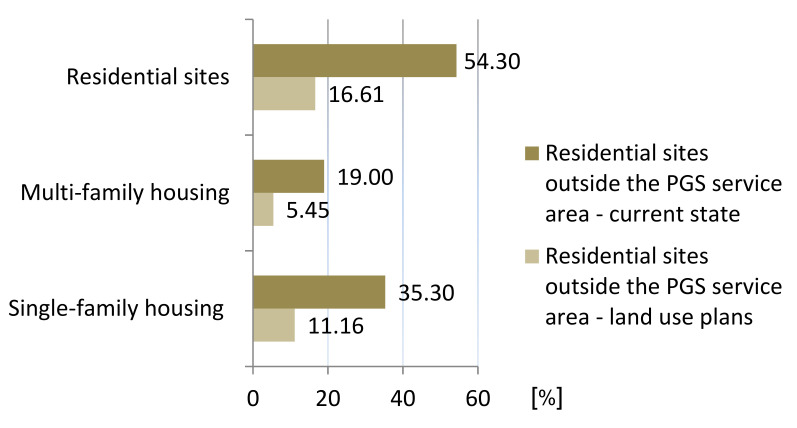
Residential sites not served by any PGS, comparison of the current state and land use plans.

**Table 1 ijerph-20-03067-t001:** Characteristics of housing estates located in the study area.

Name of the Housing Estate	Surface [ha]	Characteristic
Ołtaszyn	336.79	Incorporated into the city in 1950; the layout of the former village visible; the first contemporary complexes of buildings come from the 1970s and 1980s; dynamic development began at the end of the first decade of the 21st century.
Wojszyce	327.04	Incorporated into the city in 1950; the layout of the former village visible; the first contemporary complexes of buildings come from the 1970s and 1980s; dynamic development began at the end of the first decade of the 21st century.
Jagodno	342.71	Colony of the suburban settlement and later the town of Brochów; historic buildings of the former colony limited to a few streets. Until the 1990s the least urbanised of the surveyed areas; dynamic development began at the end of the first decade of the 21st century.

**Table 2 ijerph-20-03067-t002:** Hierarchical structure of public green spaces (PGSs) and maximum walking distances in accordance with the PGS accessibility standards proposed for Wrocław.

No	Type of Public Green Space (PGS)	Surface [ha]	Max Pedestrian Distance [m]	Short Characteristic: Functions, Facilities, Landscape Elements
1	PGS I Squares, pocket parks, mini parks	<2	400	Functions: enable passive recreation, active recreation to a limited extent and usually to a selected age group (mainly children). Facilities: paths, seats, playgrounds, and/or fitness equipment. Landscape elements: planned composition of greenery (trees, shrubs, grassy areas, and flower meadows), they may also have ornamental compositions of perennials and grasses.
2	PGS II Neighbourhood parks	2–5	800	Functions: enable active and passive recreation for all residents of a neighbourhood. Facilities: paths, seats, playgrounds, fitness equipment, areas for games, picnic, and neighbourhood cultural events. Landscape elements: planned composition of greenery (trees, shrubs, perennials, grasses, lawns, and flower meadows).
3	PGS III Supra-neighbourhood parks: district parks, city parks	>5	1200	Functions: they are diverse in nature and serve a broader purpose than the neighbourhood parks; they allow for many forms of active and passive recreation. Facilities: paths, seats, playgrounds, fitness equipment, areas for picnic, sport games and cultural events, food outlets (restaurants, cafés, and kiosks), and toilets. Landscape elements: planned composition of greenery (trees, shrubs, perennials, grasses, lawns, and flower meadows), varied topography, sculptures and elements of park architecture (e.g., pergolas, gazebos, and bridges), and water features (e.g., ponds, canals, and fountains).

**Table 3 ijerph-20-03067-t003:** Characteristics of representative PGSs from the study area and the buffer zones.

Type of PGS	No. on the Figure 4	Surface [ha]	Location, Date of Creation, Connections with Other Green Spaces	Facilities and Main Landscape Elements	Photo from the User’s Perspective and Localisation Shoved on the Orthophotomap
PGS I	8	1.09	Mini park located in the place of neglected informal green space, surrounded by new single-family housings, open in 2018 (Stage I), 2019 (Stage II).	Paths, benches, a playground, a new composition of greenery integrated into existing landscape.	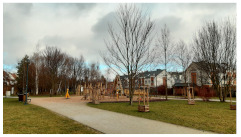 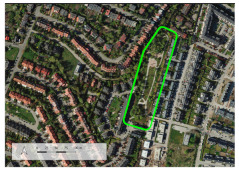
PGS II	10	2.4	Neighbourhood park located in the place of neglected informal green space; surrounded by new multi-family housings; open in 2018; connected by a green walk and a cycle path along the railway embankment with other green spaces.	Paths, benches, and a multifunctional grassy area. A new composition of greenery integrated into existing landscape.	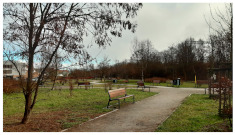 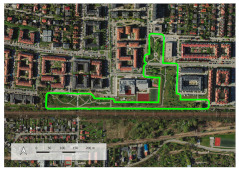
PGS III	16	23.28	A historic district park from the second part of the 19th century. Important element in an urban greenery system, connected by a historical promenade along the railway embankment with other green spaces.	Paths, benches, and lawns, large old trees and shrubs, a representative area for concerts and other cultural events, grassy areas for picnics and sport games, as well as a pond with fountains; historic elements of park architecture: a pergola, a bridge, a belvedere, a gazebo.	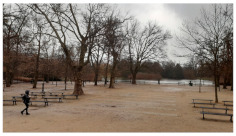 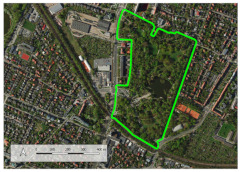

**Table 4 ijerph-20-03067-t004:** Residential sites located within the service areas of PGS and outside: current state.

	Within 400 m Service Area		Outside the 400 m Service Area		Within 800 m Service Area		Outside the 800 m Service Area		Within 1200 m Service Area		Outside the 1200 m Service Area	
	[ha]	[%]	[ha]	[%]	[ha]	[%]	[ha]	[%]	[ha]	[%]	[ha]	[%]
Single-family housing	98.00	28.87	160.00	47.13	37.96	11.20	220.00	65.00	35.8	10.54	222.2	65.51
Multi-family housing	12.68	3.73	68.16	20.27	2.71	0.80	78.13	23.00	3.03	0.9	78.81	23.06
Total	110.68	32.60	228.76	67.40	40.67	12.00	298.8	88.00	38.83	11.43	300.6	88.57

**Table 5 ijerph-20-03067-t005:** Residential sites located within the service areas of PGS and outside, according to land use plans.

	Within 400 m Service Area		Outside the 400 m Service Area		Within 800 m Service Area		Outside the 800 m Service Area		Within 1200 m Service Area		Outside the 1200 m Service Area	
	[ha]	[%]	[ha]	[%]	[ha]	[%]	[ha]	[%]	[ha]	[%]	[ha]	[%]
Single-family housing	210.95	37.90	204.70	36.85	187.9	33.84	227.80	40.90	303.80	54.68	111.90	20.14
Multi-family housing	55.36	10.00	84.52	15.21	65.76	11.86	74.12	13.34	100.10	18.00	39.80	7.16
Total	266.31	47.94	289.22	52.06	253.90	45.70	301.70	54.30	403.90	72.70	151.70	27.30

## Data Availability

Not applicable.

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
