# Peer review of "Do We Need Public Green Spaces Accessibility Standards for the Sustainable Development of Urban Settlements? The Evidence from Wrocław, Poland"

_ijerph, 2023, doi:10.3390/ijerph20043067_

Round 1
Reviewer 1 Report
The article addresses the very important issue of accessibility standards for green spaces in Poland. The city for the study was well chosen, and the issue was well presented against the background of current research in this area. The issue is new and has not been presented in this way before. Extensive and new scientific literature provides extensive and complete sources. The presented studies show the problem from Poland, the authors indicate how it can be solved using planning tools. The presented solutions can be useful where similar problems exist and there are no properly defined GPS design standards. The figures are correct and clear.
Acceptance after minor corrections and completion
Please clarify such points
- (line 13, lines 99-100) Are the standards mentioned in the summary written in any planning or directional documents of Wroclaw? Is this a proposal of the article's authors? Why these distances of 400 m, 800 m, 1200 m?
Please correct in the text
- References (line 599) - Thomson correct on Thompson
- References - in the Polish and English versions (item 74) add the name of the city for which the document was prepared
Reviewer 2 Report
The paper tackles an important topic with PGS and its distribution within cities.
The abstract needs some elaboration on the research background, problem or question, and tools with less technical language.
The paper should add a section on socio-cultural characteristics of residents of the case study, especially on the culture of using outdoor spaces.
Classifying PGS only as per their size and area could be good in an abstract context, but in a living city like the one under study, attributes related to the functions and quality of PGS could be also of importance especially for understanding people's interaction with it. even if the authors are not interested to purse such qualitative analysis, still some visual reference or characterization of the typical character if the three sizes if PGS involved in the study is particularly important to give the paper more sense of context.
Figures should be enhanced in resolution and their captions (e.g. figure 3) must be in a font a resolution to be easily readable, even if that requires expanding over more pages.
The definition of distance and reach could be better defined by network analysis than by GIS parallel buffers. Figure 6 reveals some contradiction in the logic of the analysis, with some building shown as not served on the 800 m range but served on the other two levels. This means that the analysis adopted considers that the service offered by one PGS size is not offered by the others. This in fact does not hold lots of logic when it comes to how people use PGS. in further points to the need to understand more the nature of the service being offered by these spaces and how to integrate them all in one integrated structure.
in brief the paper has two main methodological flaws:
Not clearly dealing with the overlap or functions between the varied sizes of PGS.
Missing needed explanation and elaboration on the qualitative aspects of the three levels and their respective roles within the city.
these two flaws resulted in a slim conclusion focused only on geographical/spatial analysis with little link to real life context.
Reviewer 3 Report
I view the article very positively. It is noticeable that the authors attach importance to each part of the publication, showing great efforts to ensure that they are covered comprehensively. The topic is very timely and necessary. The selection of case studies is very appropriate.
I have the following minor comments:
- please avoid citing too many publications in one place (e.g. footnote citing items 26-32);
- please further elaborate and explain to foreign readers why Wrocław as a case study fits into the international discussion;
- the authors use the phrases "local spatial development plans" and "location decisions". I have the impression that for a foreign reader these are completely unreadable. Rather (depending on the recognition of the given thematic perspective) we use "plans". "Land use plans", or "spatial plans". In contrast, "spatial development plans" in the literature are often understood rather as a form of "strategic spatial plans". I think it is worth considering modifying these terms.
